Fire severity effects on resprouting of subtropical dune thicket of the Cape Floristic Region

Strydom Tiaan 1 2 s213206781@mandela.ac.za
http://orcid.org/0000-0002-8891-2869 Kraaij Tineke 1 2
Difford Mark 2
http://orcid.org/0000-0003-3514-2685 Cowling Richard M. 2
1 School of Natural Resource Management, Nelson Mandela University , George, Western Cape , South Africa
2 African Centre for Coastal Palaeoscience, Nelson Mandela University , Port Elizabeth, Eastern Cape , South Africa
Giordani Paolo
Electronic publication date: 2020 Jun 10
Publication date: 2020
Volume: 8
Electronic Location ID: e9240
Received 2020 Jan 23; Accepted 2020 May 5
Copyright: © 2020 Strydom et al.
Copyright year: 2020
Copyright holder: Strydom et al.
License: This is an open access article distributed under the terms of the Creative Commons Attribution License, which permits unrestricted use, distribution, reproduction and adaptation in any medium and for any purpose provided that it is properly attributed. For attribution, the original author(s), title, publication source (PeerJ) and either DOI or URL of the article must be cited.
License URL: https://creativecommons.org/licenses/by/4.0/

Keywords: Biome boundaries, Coastal dunes, Fire frequency, Fire severity scoring, Pre-fire shrub size, Thicket-fynbos mosaic

Funding: National Research Foundation of South Africa 117166 Centre for Coastal Palaeoscience and the Nelson Mandela University This research was supported by the National Research Foundation of South Africa (Grant number 117166), the Centre for Coastal Palaeoscience and the Nelson Mandela University. The funders had no role in study design, data collection and analysis, decision to publish, or preparation of the manuscript.

==============================
It has been hypothesised that high-intensity fires prevent fire-dependent fynbos from being replaced by fire-avoiding subtropical thicket on dune landscapes of the Cape Floristic Region (CFR). Recent extensive fires provided an opportunity to test this hypothesis. We posit that (1) fire-related thicket shrub mortality would be size dependent, with smaller individuals suffering higher mortality than larger ones; and (2) that survival and resprouting vigour of thicket shrubs would be negatively correlated with fire severity. We assessed survival and resprouting vigour post-fire in relation to fire severity and pre-fire shrub size at two dune landscapes in the CFR. Fire severity was scored at the base of the shrub and categorised into four levels. Pre-fire size was quantified as an index of lignotuber diameter and stem count of each shrub. Resprouting vigour consisted of two variables; resprouting shoot count and resprouting canopy volume. A total of 29 species were surveyed. Post-fire survival of thicket was high (83–85%). We found that smaller shrubs did have a lower probability of post-fire survival than larger individuals but could detect no consistent relationship between shrub mortality and fire severity. Fire severity had a positive effect on resprouting shoot count but a variable effect on resprouting volume. Pre-fire size was positively related to survival and both measures of resprouting vigour. We conclude that thicket is resilient to high-severity fires but may be vulnerable to frequent fires. Prescribed high-intensity fires in dune landscapes are unlikely to reduce the extent of thicket and promote fynbos expansion.

Introduction

An understanding of the ecological determinants of the boundaries between vegetation formations (biomes) is central to managing and conserving biodiversity (Cowling & Potts, 2015; Hansen & Di Castri, 1992). South Africa is rich in biomes, the distributions of which are under a wide range of environmental and biotic controls (Bond, Midgley & Woodward, 2003; Rutherford & Westfall, 1986). A particularly interesting case is the co-occurrence of both fire-dependent (require fire for persistence) and fire-avoiding (occurring in areas were fire seldom occur) biomes in the same mesoclimate and geology, and the extent to which fire or other factors determine that co-occurrence. Notable amongst these cases is the stable co-occurrence of fire-dependent fynbos and fire-avoiding forest and thicket in South Africa’s Cape Floristic Region (CFR) (Cowling et al., 1997; Cramer et al., 2019; Geldenhuys, 1994). The persistence of fynbos species depends on regular crown fires which stimulate recruitment from persistent soil- and canopy-stored seed banks of relatively short-lived species (Pierce & Cowling, 1991). Forest and thicket, on the other hand, have predominantly short-lived, bird-dispersed diaspores which produce shade tolerant seedlings that require long fire-free intervals for recruitment into fynbos (Cowling et al., 1997; Manders & Richardson, 1992). Co-occurrence of these biomes has been attributed to differences in fire regimes (Bond, Midgley & Woodward, 2003; Geldenhuys, 1994; Manders & Richardson, 1992; Pierce & Cowling, 1991; Vlok, Euston-Brown & Cowling, 2003), soil nutrients (Coetsee, Bond & Wigley, 2015; Cramer et al., 2019; Manders & Richardson, 1992), soil moisture (Manders & Richardson, 1992) and other disturbances such as wind or large herbivores (Vlok & Euston-Brown, 2002).

The Holocene dunes of the Cape Floristic Region comprise a mosaic of dune fynbos (hereafter fynbos) and subtropical dune thicket (hereafter thicket) which co-occur in roughly equal proportions throughout the extent of this small and fragmented coastal landscape (Pierce & Cowling, 1991; Tinley, 1985; Vlok, Euston-Brown & Cowling, 2003). Casual observations suggest that most thicket species are capable of resprouting from basal and aerial bud banks but the extent to which resprouting vigour varies with fire severity (sensu Keeley, 2009) and shrub size is unknown. It has been hypothesised that fire intensity and frequency are the primary determinants of the co-occurrence and relative abundances in dune landscapes of these two biomes; occasional high-intensity burns destroy mature thicket shrubs whereas frequent fires kill young thicket plants that have established in fynbos (Cowling et al., 1997; Cowling & Potts, 2015; Vlok, Euston-Brown & Cowling, 2003). However, other than some observations of post-fire regeneration in fynbos (Cowling & Pierce, 1988), there has been little research on the fire ecology of dune thicket communities and nothing is known about the response of thicket shrubs to different fire regimes. Generally, the fine fynbos fuels are more flammable and burn at higher intensities than coarser thicket fuels (Burger & Bond, 2015; Calitz, Potts & Cowling, 2015), except under extreme conditions when thicket can burn at intensities that may exceed those in fynbos (Kraaij et al., 2018). In fynbos ecosystems, high-intensity fires promote good regeneration of obligate reseeding (non-sprouting) species (Bond, Le Roux & Erntzen, 1990), whereas low-intensity fires benefit resprouting species (Vlok & Yeaton, 1999; Vlok & Yeaton, 2000). Under natural conditions fire return intervals in dune landscapes are thought to be anything from 16 to 26 years in fynbos and 50 years or more in unbroken stands of thicket (Cowling et al., 1997; Kraaij et al., 2013a; Cowling & Potts, 2015).

An important aim of our study was to provide recommendations for the ecological management of fire in dune landscapes that will ensure co-occurrence of its two component biomes. Fire suppression has long been practised in dune landscapes in order to protect assets (particularly coastal resort developments and exotic timber plantations), which has resulted in near absence of fires for decades and the accumulation of large fuel loads (Cowling et al., 1997; Kraaij, Cowling & Van Wilgen, 2011). Prescribed burning for ecological purposes is difficult to implement as fires in long-unburnt vegetation pose significant safety risks (Van Wilgen, 2009). Should managers implement frequent, low-intensity fires to reduce fuel accumulation or should they implement infrequent high-intensity fires to reduce thicket extent and encroachment? Recent extensive fires along the southeastern CFR coast (Kraaij et al., 2018) presented an opportunity to assess the response of thicket shrubs to varying fire severity. In particular, we set out to test the following hypotheses: (1) fire-related thicket shrub mortality would be size dependent, with smaller individuals suffering higher mortality than larger ones; and (2) survival and resprouting vigour of thicket shrubs would be negatively correlated with fire severity. We tested these hypotheses by surveying post-fire survival and resprouting vigour of thicket shrubs in relation to fire severity and pre-fire shrub size. We surveyed two sites (176 km apart) that burnt at different times (January 2016 and June 2017) and under different weather and climatic conditions. This provided us with a wide range of fire intensities (within and between fires) that allowed us to get a sense of the geographic generality of any findings.

Materials and Methods

Study area

The study area comprises of two dune landscape sites (hereafter ‘sites’) along the southeastern CFR coast of South Africa, where fires have occurred in recent years, that is at Cape St. Francis (34.19028° S 24.82112° E) and Knysna (34, 24046° S 22, 50438° E) (Fig. 1). Permission was obtained from CapeNature (permit number CN35-59-4006) to work on land under their management. Permission was also obtained from private landowners Christa le Roux, Jan Rigaard and Susan Cambell. The Cape St. Francis fire was 1,000 ha in extent, occurred in summer (25 January–1 Febuary 2016), and followed a year of average rainfall (Fig. S1). The mean fire danger rating during the month preceding the fire was dangerous, whereas those on the days of the fire varied from moderate to extremely dangerous (Table S1). The Knysna fire was 15,000 ha in extent (of which approximately 2,500 ha comprised Holocene dunes with thicket-fynbos mosaic vegetation), occurred in winter (6–11 June 2017), and followed the worst 18-month drought in recorded history (Kraaij et al., 2018; Fig. S1). The mean fire danger rating during the month preceding the fire was dangerous, whereas those on the days of the fire varied from moderate to extremely dangerous (Table S1). The pre-fire age of the vegetation was unknown at both Cape St. Francis and Knysna but was deemed to be in excess of 40 years at both sites. The soils at these sites comprise deep, well-drained, calcareous sands associated with hairpin (east-west-trending) dunes of terminal Pleistocene to Holocene origin (Tinley, 1985). Annual precipitation varies between 600 and 900 mm and occurs throughout the year with slight peaks in autumn and spring (Rebelo et al., 2006). Fire may occur at any time of the year with fire danger weather peaking in the dry summer months and in autumn and winter when warm and dry katabatic or ‘berg’ winds are common (Kraaij, Cowling & Van Wilgen, 2013).

Figure 1 The two study sites, Knysna and Cape St. Francis, that burnt and the extent of Holocene dunes along the southeast coast of the Cape Floristic Region.

The two smaller maps show the extent (coloured in red) of the June 2017 fires at Knysna and January 2016 fire at Cape St. Francis. Imagery source: Estri, DigitalGlobe, GeoEye, Earthstar Geographics, CNES/Airbus DS, USDA, USGS, AeroGRID, IGN. Distribution data for Holocene dunes were derived from the Council for Geoscience 1:250,000 geological database.

According to the national classification, the vegetation of the area is Southern Cape Dune Fynbos (Rebelo et al., 2006). In particular, the calcareous dunes of the CFR’s southeast coast support mosaics of fynbos with floristic affinities to the warm-temperate CFR and thicket of subtropical affinity (Pierce & Cowling, 1991; Vlok, Euston-Brown & Cowling, 2003) (Fig. S2). In areas protected from fire, such as narrow dune valleys with steep-sided walls, dune thicket reaches forest stature and includes several species that are rare in fire-exposed thicket for example Apodytes dimidiata (Icacinaceae), Chionathus foveolatus (Oleaceae), Zanthoxylum capense (Rutaceae) (Cowling et al., 1997; Van der Merwe, 1976). Thicket is dominated by tall (2–5 m), mesophyllous shrubs that form a dense, closed and often impenetrable shrubland (Vlok, Euston-Brown & Cowling, 2003) and is regarded as a fire-avoiding biome (Cowling et al., 2005; Linder, 2005). Fynbos, on the other hand, seldom exceeds 2 m in height and is a much more open community. Thicket shrubs produce short-lived, bird-dispersed fruits and recruitment typically occurs in shaded microsites during the inter-fire stage (Cowling et al., 1997). Most dune thicket shrubs appear capable of basal sprouting as multi-stemmed individuals are common (Kruger, Midgley & Cowling, 1997; Midgley & Cowling, 1993); sprouting from aerial shoots is also a feature of many species (T. Strydom, 2018, personal observation). All our sample sites were located in burnt thicket, but some sites did include populations of forest species, especially those of taller (>5 m) stature.

Data collection

Field surveys were undertaken during April 2018 (28 months post-fire) at Cape St. Francis and during August 2018 (14 months post-fire) at Knysna. Belt transects (n = 51) of 50 m × 5 m were positioned within the burnt areas to incorporate various fire intensities and the thicket shrub species composition and abundance that were representative of the area. We sampled 90% of the species of canopy forming thicket shrubs that occur at the Knysna site and all such shrubs at the Cape St. Francis site. Plant identification was based on the leaves of the resprouts of live individuals and the shrub architecture and bark texture of individuals killed by fire. Within transects, we measured shrubs with a lignotuber diameter of >5 cm to avoid missing small shrubs that may have burnt away in, or decayed after, the fire. An individual shrub was considered to be all stems radiating out from a central lignotuber, with these stems no further than 5 cm away from the lignotuber (Marais et al., 2014). Measurements included the fire severity (‘Firebase’) an individual shrub has experienced, its pre-fire dimensions (that of the burnt skeleton) and post-fire resprouting vigour. Fire severity was assessed by subjectively scoring the extent of damage to the base of the stem (at 0–50 cm from the ground) on each shrub as follows: 1 = low (bark burnt black, but intact); 2 = medium (considerable fire damage to bark); 3 = high (bark destroyed by fire and considerable damage to the wood); 4 = extreme (extreme damage to the wood, which is partially consumed by fire) (Fig. S3). We also scored fire severity in terms of damage to the canopy and size of remaining material after established methods of Moreno & Oechel (1991) and Marais et al. (2014) and found that fire severity scores at the base and canopy levels were significantly correlated (rs = 0.77; p < 0.001). We therefore only used fire severity at the base for subsequent analyses as this measure is most relevant to plants, such as our study species, that display basal sprouting. Pre-fire dimensions (regarded as proxies for pre-fire biomass) were assessed by measuring the lignotuber diameter at the base and counting the number of pre-fire stems as a measure of stored reserves, available bud bank and age (Marais et al., 2014). Post-fire resprouting vigour was assessed in terms of (i) resprouting canopy diameter (the mean of the longest and shortest diameter), (ii) the number of shoots resprouting from the shrub base (no further than 5 cm away from the lignotuber); (iii) length (cm) of the longest resprouting shoot, and (iv) variability in the lengths of resprouting shoots by allocating a subjective score of low variability (all shoots approximately the same length), medium variability (noticeable difference in shoot lengths) or high variability (high variability in shoot lengths). We aimed to survey a minimum of 30 individuals of each species, but this could not be achieved for all species as some species were naturally rare.

Data analysis

The unit of replication comprised of individual shrubs (species were assessed jointly and included as a random factor); sample sizes at Cape St. Francis (519 shrubs across 31 transects) and Knysna (593 shrubs across 20 transects) were comparable. We compared the median fire severity score between the two sites using a Mann–Whitney U test. We assessed survival of all thicket shrubs from both study sites using a logistic regression model (binomial family, logit link) from the open source R software (version 3.6.3) (R Development Core Team, 2019) (Code S1; Table S2). As predictors of survival we included fire severity (an ordered factor with four levels), an index of pre-fire size (see below), site (a two-level factor), and all interactions between the predictors. Pre-fire shoot count and pre-fire lignotuber diameter (both measures of pre-fire size) were highly correlated and, therefore, were combined into an index of pre-fire size by calculating as the sum of pre-fire stem count and pre-fire lignotuber diameter, each of which was transformed to range between 0 and 1 (to weight both measures equally). We tested for collinearity between fire severity and pre-fire size, but this relationship was weak (Fig. S4) and both factors were therefore retained in analyses. We employed stepwise selection to identify the best subset model using AIC as the model selection criterion (Akaike, 1974; Symonds & Moussalli, 2011). Factors included in the final selected model were fire severity, pre-fire size, site, and the two-way interaction between fire severity and pre-fire size.

We assessed the number of resprouting shoots (an integer-valued count variable) occurring at the shrub base of all thicket shrubs using the Delaporte distribution for count data. This is a three-parameter distribution available in the gamlss (generalized additive models for location, scale and shape) contributed package for R (Stasinopoulos & Rigby, 2007). We modelled the mean using the default logarithmic link, and dispersion and skewness using null models (logarithmic links). As predictors, we included fire severity (a four-level ordered factor), pre-fire size (the index detailed above), site (a two-level factor), and all interactions as fixed factors (Code S2; Table S2). We employed stepwise selection to identify the best subset model using AIC as the model selection criterion (Akaike, 1974; Symonds & Moussalli, 2011). The full model was selected as the final model.

We used a two-parameter Weibull distribution (Stasinopoulos & Rigby, 2007) to assess the resprouting volume of all thicket shrubs in relation to fire severity (an order factor), pre-fire size (the index detailed above), site (a two-level factor), and all interactions as fixed factors (Code S3; Table S2). For the response variable, resprouting volume, we estimated canopy volume as the volume of a cone (Volume=1/3πr2h), where r represents the mean of the long and short diameter of the resprouting canopy divided by two, and h represents the length of the longest resprouting shoot. We adjusted the length of the longest resprouting shoot by a factor of 0.75 for a shrub that displayed medium variability in resprouting shoot length and by a factor of 0.50 for a shrub displaying high variability in shoot length, and no adjustment for a shrub displaying low variability in shoot length. The model was fitted using the default logarithmic link function, coupled with the default null model for dispersion, and fitted using a logarithmic link function. We used AIC-based stepwise selection to identify the best subset model (Akaike, 1974; Symonds & Moussalli, 2011). Factors included in the final selected model were fire severity, pre-fire size, site, and the interactions between fire severity and site, and pre-fire size and site.

Results

The median (±SE) fire severity score was significantly higher (Z1.92 = 10.54, P < 0.000) at Cape St. Francis (4 ± 0.9) than at Knysna (3 ± 0.9). A total of 29 species were surveyed, three of which occurred only at Cape St. Francis and 12 of which occurred only at Knysna (Table 1). All species surveyed were capable of resprouting after fire, with survival being very high at Cape St. Francis (85% of individual shrubs) and Knysna (83%) (Table 1). Resprouting was virtually exclusively from the shrub bases, but a few species showed epicormic resprouting when fire severity in the canopy was low, including Chionanthus foveolatus (Oleaceae), Pterocelastrus tricuspidatus (Celastraceae), Sideroxylon inerme (Sapotaceae) and Tarchonanthus littoralis (Asteraceae) but we suspect all the species would resprout epicormically after low severity fire. Species showing lowest survival were Psydrax obovata (Rubiaceae), Scolopia zeyheri (Flacourtiaceae), Apodytes dimidiata (Icacinaceae), Gymnosporia buxifolia (Celastraceae), Sideroxylon inerme and Chionanthus foveolatus. All the final models assessing the effects on survival, post-fire resprouting shoot count and resprouting volume of fire severity, pre-fire size, site, and their interactions, explained only small proportions of the overall deviance (r2 = 0.05, 0.09 and 0.19, respectively; Tables 2–4), suggesting that most variation was not accounted for by the factors investigated. Amongst these factors, pre-fire size was the primary determinant of the post-fire response, showing consistently positive effects on survival, post-fire resprouting shoot count and post-fire resprouting volume (Figs. 2–4; Tables 2–4). Fire severity and site had comparatively minor effects (in terms of the amount of deviance explained, Tables 2–4) and often interacted with each other or with pre-fire size. Shrubs generally survived better and resprouted more vigorously at Cape St. Francis than they did at Knysna, despite fire severity being higher at Cape St. Francis. The effect of site on survival and post-fire resprouting shoot count was particularly small (explaining little deviance relative to the other factors; Tables 2–4) but was more important on post-fire resprouting volume (explaining a similar amount of deviance to fire severity). Fire severity interacted with pre-fire size in its effect on survival (Table 2), with high fire severity possibly having more pronounced negative effects on the survival of large shrubs (Fig. 2). Fire severity had a generally positive effect on post-fire resprouting shoot count and interacted with site and pre-fire size in complex ways (all second and third-order interactions were significant) (Fig. 3; Table 3). Site interacted with pre-fire size and with fire severity in its effects on post-fire resprouting volume (Table 4). The positive effect of pre-fire size on post-fire resprouting volume was more pronounced at Knysna than it was at Cape St. Francis, while fire severity had opposing effects on post-fire resprouting volume at the two sites (Fig. 4).

Table 1 Percentage survival, mean pre-fire size and sample number of thicket species surveyed at Cape St. Francis (CSF) and Knysna (KNS).

Pre-fire size comprises an index calculated as the sum of pre-fire shrub lignotuber diameter and pre-fire stem count after both measures have been transformed to range between 0 and 1.

Species	Survival (%)	Mean
pre-fire size	Sample number	
CSF	KNS	CSF	KNS	CSF	KNS	
Acokanthera oppositifolia (Lam.) Codd	100	100	–	0.26	–	1	
Allophylus decipiens (Sond.) Radlk.	–	92	–	0.17	–	13	
Apodytes dimidiata E.Mey. ex Am.	–	59	–	0.21	–	56	
Buddleja saligna (Wild.)	–	100		0.29	–	8	
Cassine peragua L.	100	79	0.41	0.36	21	58	
Chionanthus foveolatus (E.Mey.) Stearn	–	79	–	0.20	–	30	
Clausena anisata (Wild.) Hook.f. ex Benth.	100	100	0.25	0.23	15	11	
Diospyros dichrophylla (Grand.) De Winter	–	84	–	0.21	–	37	
Dovyalis rhamnoides (Burch. ex DC.) Burch.and Harv.	100	100	0.31	0.11	16	2	
Dovyalis rotundifolia (Thunb.) Thunb. and Harv.	100	–	0.30	–	14	–	
Euclea racemosa L.	92	92	0.30	0.21	39	13	
Gymnosporia buxifolia (L.) Szyszyl.	100	60	0.34	0.26	6	5	
Gymnosporia nemorosa (Eckl. and Zeyh.) Szyszyl.	–	90	–	0.32	–	12	
Halleria lucida L.	–	100	–	0.17	–	2	
Mystroxylon aethiopicum (Thunb.) Loes.	86	100	0.54	0.29	97	10	
Olea europaea subsp. africana L.	–	82	–	0.26	–	45	
Olea exasperata Jacq.	100	100	0.21	0.17	2	11	
Osyris compressa (P.J.Bergius)	100	–	0.62	–	2	–	
Pittosporum viridiflorum Sims	–	100	–	0.29	–	5	
Psydrax obovata (Eckl. and Zeyh.) Bridson	40	–	0.50	–	5	–	
Pterocelastrus tricuspidatus (Lam.) Walp.	90	80	0.80	0.42	136	141	
Rapanaea melanophloeos (L.) Mez	–	100	–	0.17	–	7	
Scolopia zeyheri (Nees). Harv.	100	51	0.25	0.22	2	39	
Scutia myrtina (Burm.f.) Kurz	100	100	0.31	0.14	28	3	
Searsia glauca (Thunb.) Moffett	81	80	0.46	0.24	108	20	
Searsia lucida (L.) F.A Barkley	100	87	0.50	0.31	49	55	
Sideroxylon inerme L.	76	71	0.51	0.44	97	78	
Tarchonanthus littoralis P.P.J. Herman	–	100	–	0.37	–	80	
Zanthoxylum capense (Thunb.) Harv.	79	83	0.42	0.13	14	6	
All species collectively	85	83	0.41	0.51	519	593	

Table 2 Output of a logistic regression model (binomial family, logit link function) investigating the effects of fire severity, pre-fire size and site on post-fire thicket shrub survival.

Effects include fire severity scored at the base of each shrub (Firebase; orthogonal polynomial contrast trend test; L, linear; Q, quadratic; C, cubic), pre-fire size (Prefiresize; see Methods for details), site and fire severity × pre-fire size interaction. Models were fitted using treatment contrast with Cape St. Francis as the reference level for site comparison with Knysna (KNS).

Survival	Regression coefficients are log odds, standard errors in brackets	
(Intercept)	1.28 (0.23)***	
Firebase.L	0.23 (0.46)	
Firebase.Q	0.66 (0.38)	
Firebase.C	0.88 (0.27)**	
Prefiresize	2.52 (0.68)***	
SiteKNS	−0.37 (0.17)*	
Firebase.L:Prefiresize	−2.98 (1.68)	
Firebase.Q:Prefiresize	−0.58 (1.36)	
Firebase.C:Prefiresize	−2.08 (0.94)*	
Number of observations	1,398	
Nagelkerke R2	0.05	
Notes:

* p < 0.05.

** p < 0.01.

*** p < 0.001.

Table 3 Output of a Delaporte count model investigating the effects of fire severity, pre-fire size and site on thicket shrub resprouting shoot count.

Effects include fire severity scored at the base of each shrub (Firebase; orthogonal polynomial contrast trend test; L, linear; Q, quadratic; C, cubic), pre-fire size (Prefiresize), site, fire severity × pre-fire size interaction, fire severity × site interaction, pre-fire size × site interaction and fire severity × pre-fire size × site interactions. Coefficients for sigma and nu are for dispersion (variance), and skewness. Models were fitted using treatment contrasts with Cape St. Francis as the reference level for Site comparison with Knysna (KNS).

Resprouting shoot count	Regression coefficients are natural logarithms, standard errors are in brackets	
(Intercept)	2.38 (0.09)***	
Firebase.L	0.17 (0.21)	
Firebase.Q	−0.41 (0.18)*	
Firebase.C	0.03 (0.14)	
Prefiresize	0.40 (0.11)***	
SiteKNS	−0.26 (0.12)*	
Firebase.L:Prefiresize:	0.31 (0.25)	
Firebase.Q:Prefiresize:	0.50 (0.22)*	
Firebase.C:Prefiresize:	−0.02 (0.19)	
Prefiresize:SiteKNS	0.72 (0.19)***	
Firebase.L:SiteKNS	−0.26 (0.26)	
Firebase.Q:SiteKNS	0.81 (0.23)***	
Firebase.C:SiteKNS	−0.35 (0.19)	
Firebase.L:Prefiresize:SiteKNS	0.10 (0.44)	
Firebase.Q:Prefiresize:SiteKNS	−1.25 (0.39)**	
Firebase.C:Prefiresize:SiteKNS	0.42 (0.34)	
sigma (Intercept)	0.20 (0.10)*	
nu (Intercept)	−1.36 (0.13)***	
Number of observations	1112	
Nagelkerke R2	0.09	
Notes:

* p < 0.05.

** p < 0.01.

*** p < 0.001.

Table 4 Output of a Weibull model investigating the effects of fire severity, pre-fire size and site on post-fire resprouting volume.

Effects include fire severity scored at the base of each shrub (Firebase; orthogonal polynomial contrast trend test; L, linear; Q, quadratic; C, cubic), pre-fire size (Prefiresize), site, fire severity × site interaction and pre-fire size × site interaction. Sigma represents the coefficient for dispersion (variance). Models were fitted using treatment contrasts with Cape St. Francis as the reference level for Site comparison with Knysna (KNS).

Resprouting volume	Regression coefficients are natural logarithms, standard errors are in brackets	
(Intercept)	12.61 (0.16)***	
Firebase.L	0.58 (0.22)**	
Firebase.Q	0.09 (0.18)	
Firebase.C	−0.09 (0.16)	
Prefiresize	0.91 (0.22)***	
SiteKNS	−1.03 (0.21)***	
Firebase.L:SiteKNS	−1.91 (0.27)***	
Firebase.Q:SiteKNS	0.15 (0.23)	
Firebase.C:SiteKNS	0.18 (0.20)	
Prefiresize:SiteKNS	1.65 (0.38)***	
sigma (Intercept)	−0.47 (0.02)***	
Num. obs.	1,112	
Nagelkerke R2	0.19	
Notes:

* p < 0.05.

** p < 0.01.

*** p < 0.001.

Figure 2 Predicted probability of post-fire survival of dune thicket shrubs in relation to the interaction between fire severity and pre-fire shrub size.

Fire severity was scored at the base of each shrub; Firebase (A) 1, low; (B) 2, medium; (C) 3, high; (D) 4, extreme. Pre-fire shrub size is an index (Prefiresize; see “Methods” for details). The output of the logistic regression model that was used to calculate the effects is given in Table 2.

Figure 3 Predicted effects of the interaction between fire severity, pre-fire shrub size and site on the logarithm of the post-fire shoot count of dune thicket shrubs.

Fire severity was scored at the base of each shrub (Firebase = 1, low; 2, medium; 3, high; 4, extreme). Pre-fire shrub size is an index (Prefiresize; see “Methods” for details). Sites were (A–D) CSF, Cape St. Francis; and (E–H) KNS, Knysna. The output of the Delaporte count model that was used to calculate the effects is given in Table 3.

Figure 4 Predicted effects of the interactions between (A and B) pre-fire size and site, and (C and D) fire severity and site on the logarithm of post-fire resprouting volume of dune thicket shrubs.

Fire severity was scored at the base of each shrub (Firebase = 1, low; 2, medium; 3, high; 4, extreme). Pre-fire shrub size is an index (Prefiresize; see “Methods” for details). The output of the Weibull model that was used to calculate the effects is given in Table 4.

Discussion

Importance of fire in maintaining boundaries between thicket and fynbos

Contrary to tacit understanding (Cowling et al., 1997; Cowling & Potts, 2015; Vlok, Euston-Brown & Cowling, 2003), our results showed that fire severity has only minor or inconsistent (among sites and shrub size classes) effects on the survival and resprouting vigour of thicket shrubs. We assessed the community level response of thicket to fire, by considering species as a random factor and including shrub species composition that was representative of the communities at our study sites. The fires assessed in the current study were of high to extreme severity (based on our assessments and other evidence, Kraaij et al., 2018; Swana, 2016) and despite this, survival of thicket shrubs was remarkably high. Only the very largest of shrubs (basal diameter 80–100 cm; Cassine peragua, Pterocelastrus tricuspidatus, Sideroxylon inerme) exhibited an increased susceptibility to mortality at the highest fire intensities but the probability of survival in these large shrubs was still above 80% (Fig. 2). Equally the lowest survival probability (i.e. under high fire severity) exhibited by the smallest shrubs was 60 %. Bud protection and bark production (Clarke et al., 2013; Charles-Dominique et al., 2015) (which was not investigated) may be an important factor explaining variation in survival and resprouting vigour of thicket shrubs post-fire as this was shown to be important in savanna trees (Charles-Dominique et al., 2015). Fire severity is, therefore, not deemed a major driver of mortality of thicket shrubs and hence is unlikely to play an important role in maintaining the boundaries and stable co-occurrence of thicket and fynbos in coastal dune landscapes.

The pre-fire size of shrubs was the most significant predictor of post-fire survival and resprouting vigour of thicket, with bigger shrubs (stem diameter >10 cm) showing improved survival and resprouting vigour than smaller shrubs (stem diameter ≤ 10 cm). This suggest that longer fire cycles would enable thicket shrubs to achieve sizes that conferred fire survival and thus promote thicket development. In many fire-prone ecosystems increased fire frequency has significant effects on vegetation structure and composition (Hoffmann & Solbrig, 2003; Kraaij et al., 2013b; Roques, O’Connor & Watkinson, 2001; Van Wilgen, Richardson & Seydack, 1994), through keeping woody plants small and therefore vulnerable to fire (Bond & Midgley, 2001). In savannas, very short (1–6 year) fire return intervals (Govender, Trollope & Van Wilgen, 2006) reduce the tree-grass ratio by supressing post-fire recovery of some woody species despite the ability to resprout (Bond & Midgley, 2001; Hoffmann & Solbrig, 2003; Roques, O’Connor & Watkinson, 2001; Smit et al., 2010). For instance, in southeastern Australian savanna where fire is deemed an important factor to maintain tree-grass co-existence, only 20–30% of small trees survived high intensity fire (Morrison & Renwick, 2000). In our study, smaller thicket shrubs generally showed poorer survival and resprouting vigour than larger shrubs, although the average post-fire survival of small thicket shrubs was high (>80%) across fire severities. This suggests that frequent fires would maintain smaller thicket shrubs that are more prone to suffer supressed recovery potential than larger shrubs, but that the effects of frequent fires may be limited.

In our study, site had a relatively minor effect on the post-fire response in terms of survival and number of resprouting shoots, but a more important effect on resprouting canopy volume. The latter may be primarily explained by the post-fire age difference at the time of surveying between the two sites. Cape St. Francis was surveyed at 38 months post-fire as opposed to 14 months post-fire at Knysna, which meant that resprouting canopies would have been larger at the time of surveying at Cape St. Francis. Given the difference in post-fire age between the two sites, apical dominance (Aarssen, 1995) could also have had disparate effects at the two sites, supported by our finding that the post-fire shoot count was generally lower at Cape St. Francis than at Knysna (Fig. 3). Another potential contributing factor may be bud tissue damage after high severity fire, resulting in less carbohydrate reserves available to resprouting shoots and thus reducing resprouting vigour (Clarke et al., 2013). The mean fire severity score at Cape St. Francis was higher than at Knysna, suggesting that bud tissue damage should have been more extensive at the former, yet shrubs at Cape St. Francis displayed greater resprouting volume. Potential effects of pre-fire reserves are however clouded by disparate post-fire ages at the time of measurement. We thus contend that a combination of post-fire age and apical dominance largely explains the difference in resprouting vigour between the two sites. Discrepancies in post-fire rainfall are unlikely to have played a large role as both sites received somewhat less than the long-term mean annual rainfall in the post-fire period (Fig. S1). Resprouting shrubs are furthermore known to be less sensitive to drought conditions after fire than reseeding shrubs (Zeppel et al., 2015). However, the severe drought that prevailed at the Knysna site pre-fire (Kraaij et al., 2018; Fig. S1) could have contributed to depressed resprouting vigour compared to that at Cape St. Francis. Since site did not have strong or consistent effects on the post-fire response, this suggests that the observed vigour in post-fire recovery of thicket shrubs is likely to be generic across fires of varying intensity, season and size. Detailed investigations at species level were beyond the scope of this study, but survival rates and resprouting vigour varied considerably among species (Table 1). Jointly, the factors investigated in our models did not explain a large part of the variation in the post-fire response of thicket shrubs. We therefore suggest that species, shrub architecture, and local site effects warrant further investigation.

Other factors that may sustain thicket-fynbos boundaries

Various other factors may potentially be of importance in determining the boundaries between thicket and fynbos in dune landscapes. Cramer et al. (2019) showed that differences in soil nutrients largely account for the boundaries between forest and fynbos. Here fire and soil nutrients interact, with frequent fires killing forest seedlings and consuming the organic components of mulch-rich top-soils, thereby preventing nutrient-demanding forest species from invading nutrient-poor fynbos soils maintained by recurrent fire (Coetsee, Bond & Wigley, 2015; Cramer et al., 2019; Manders & Richardson, 1992). On the contrary, unlike the case on other substrata (Cowling & Potts, 2015), soil nutrient levels (including organic carbon and nitrogen) in our dune system are comparable between thicket and fynbos (Table S3; Cowling, 1984). We speculate that soil moisture may be of greater importance in explaining the boundaries between these two biomes in dune landscapes. Manders & Richardson (1992) suggested that soil nutrients together with soil moisture are important in sustaining the boundaries between forest and fynbos. Thicket grows predominantly at the base of dunes and in well-drained dune swales (Cowling, 1984; De Villiers et al., 2005); moisture levels in these sites are invariably higher than dune crests and the upper slopes (Maun, 2009; Tinley, 1985) where fynbos dominates. Although a subset of thicket shrubs does grow in these drier sites, they remain sparse, stunted and sub-dominant to the fynbos component (Cowling, 1984).

Megaherbivores such as elephant and black rhinoceros browsed in dune landscapes prior to their extirpation during colonial times (Boshoff & Kerley, 2001; De Villiers et al., 2005; Radloff, 2008). While these herbivores do not collapse the structure of thicket where it exists in extensive patches (Cowling et al., 2009; Radloff, 2008), their impacts where thicket forms a mosaic with fire-prone shrublands, have not been studied. They may have opened up thicket canopies, enabling shade intolerant and highly flammable fynbos shrubs to invade. However, as we have shown here, most thicket species’ post-fire survival is weakly influenced by fire severity. Furthermore, megaherbivore impacts would not explain the absence of thicket from steep dunes slopes which would have been avoided by these bulky browsers (Cowling et al., 2009).

Management implications

Our study demonstrated that all the thicket shrub species surveyed are capable of surviving fire and, moreover, fire severity had an unexpected effect on resprouting vigour in the form of increased number of resprouting shoots. Therefore, high intensity prescribed burning of dune landscapes does not offer an effective means of limiting thicket shrub encroachment and densification in fynbos. Although thicket does invade fynbos when the conditions are suitable (Cowling et al., 1997), we suspect it would not invade all of the fynbos due to other drivers, principally soil moisture availability, that maintain the boundaries between these two biomes. However, frequent fires may prevent the encroachment of thicket juveniles into fynbos, owing to the greater vulnerability of small shrubs to fire. Most recruitment of thicket individuals into fynbos is via ramets (Midgley & Cowling, 1993) that likely originate from well-established plants in mature thicket clumps (Cowling, 1984). The extent to which the development of these ramets into taller individuals that can overtop fynbos is controlled by fire or resource availability is unknown. Nonetheless, it is clear from our study that high-intensity fires are not an effective tool for limiting thicket extent in the dune landscapes of the southeastern CFR. This is good news for managers, since these fires are difficult to control and thus represent a risk to infrastructure. Instead, we recommend that frequent (5–10 year intervals), medium intensity burns be explored as a potential means to prevent thicket encroaching into fynbos areas and facilitate persistence of fire-dependent fynbos communities. In contrast to thicket which we have shown to comprise virtually exclusively of resprouting shrub species, a large component (38%) of dune fynbos shrub species are non-sprouters (obligate reseeders) while the remainder are facultative resprouters (Cowling et al., 2019). Reseeding fynbos species require medium to high intensity fires at intervals exceeding the juvenile period of obligate reseeders to recruit prolifically after fire (Kraaij & Van Wilgen, 2014). While too frequent fires may prevent seedling recruitment of slow-maturing obligate reseeders in mountain fynbos (Kraaij et al., 2013b), dune fynbos shrubs are short-lived with short juvenile periods (lifespan of 5–50 years; Pierce & Cowling, 1991). On these grounds and supported by the model predictions of Cowling et al. (1997) we suggested a regime of frequent and medium intensity fires, although the impacts of such a fire regime on dune fynbos-thicket dynamics require further research.

Conclusion

Our findings based on two distinct fire events confirm that fire severity is not an important determinant of thicket shrub survival and ultimately does not offer a management tool to limit thicket encroachment and densification into fynbos. However, pre-fire size of the thicket shrubs had consistent positive effects on post-fire survival and resprouting vigour, with larger shrubs being more resilient. We suggest there are many other factors that are of greater importance in determining these biome boundaries in dune landscapes.

Supplemental Information

Supplemental Information 1 Comparison of pre-fire and post-fire rainfall with long-term rainfall at Cape St Francis and Knysna.

Click here for additional data file.

Supplemental Information 2 Thicket and fynbos mosaics at the two study sites: (A) Cape St. Francis and (B) Knysna. Resprouting of common thicket species: (C) Pterocelastrus tricuspidatus and (D) Sideroxylon inerme.

Click here for additional data file.

Supplemental Information 3 Fire severity scoring examples for thicket shrubs: 1, low severity; 2, medium severity; 3, high severity; 4, extreme severity.

Click here for additional data file.

Supplemental Information 4 Correlation between fire severity score (firebase) and pre-fire size (an index of lignotuber diameter and stem count) of each shrub.

Click here for additional data file.

Supplemental Information 5 The Lowveld Fire Danger Index (FDI-W)A,B,C prior to and days of fire (shaded in grey) for Cape St. FrancisD and Knysna.

Click here for additional data file.

Supplemental Information 6 Analysis of Deviance Table (Type II tests) (summarised in Tables 2, 3 and 4) for survival, post-fire resprouting shoot count and post-fire resprouting volume.

Click here for additional data file.

Supplemental Information 7 Soil nutrient differences between dune fynbos, thicket and forest (Cowling, 1984).

Click here for additional data file.

Supplemental Information 8 Code used in R (version 1.1.383) (R Development Core Team, 2013) to assess survival for dune thicket shrubs.

Survival = logistic regression model (binomial family and logit link function).

Click here for additional data file.

Supplemental Information 9 Code used in R (version 1.1383) (R Development Core Team, 2013) to assess post-fire resprouting shoot count for dune thicket shrubs.

Resprouting shoot count = generalised additive model for location, scale and shape (Delaporte distribution, logarithmic link function).

Click here for additional data file.

Supplemental Information 10 Code used in R (version 1.1.383) (R Development Core Team, 2013) to assess post-fire resprouting volume for dune thicket shrubs.

Resprouting volume = generalised additive model for location, scale and shape (Weibull distribution, logarithmic link function).

Click here for additional data file.

Supplemental Information 11 Data used in analysis of resprouting vigour for thicket shrubs.

Click here for additional data file.

Supplemental Information 12 Data used in the analysis of survival for thicket shrub.

Click here for additional data file.

We thank Zanri Schoeman, Samukelisiwe Msweli and Thomas Vos whom assisted with field surveys. We thank Adriaan Grobelaar for producing a map of the study sites. Rainfall and fire danger weather data were supplied by the South African Weather Service. Reviewers provided useful suggestions that led to improvements to the manuscript.

Additional Information and Declarations

Competing Interests

Author Contributions

Field Study Permissions

Data Availability

Richard M. Cowling and Tineke Kraaij are Academic Editors for PeerJ.

Tiaan Strydom performed the experiments, analysed the data, prepared figures and/or tables, authored or reviewed drafts of the paper, and approved the final draft.

Tineke Kraaij conceived and designed the experiments, authored or reviewed drafts of the paper, and approved the final draft.

Mark Difford analysed the data, prepared figures and/or tables, authored or reviewed drafts of the paper, and approved the final draft.

Richard M. Cowling conceived and designed the experiments, authored or reviewed drafts of the paper, and approved the final draft.

The following information was supplied relating to field study approvals (i.e., approving body and any reference numbers):

Permission was obtained from CapeNature to conduct surveys for research purposes on land under their management (research permit number: CN35-59-4006). Christa le Roux, Jan Rigaard and Susan Campbell granted permission to conduct surveys on their private land.

The following information was supplied regarding data availability:

The raw data are available in the Supplemental Files.

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
