# Peer review of "Fire severity effects on resprouting of subtropical dune thicket of the Cape Floristic Region"

_PeerJ, doi:10.7717/peerj.9240_

## Round 0.1 · original submission · Minor Revisions

All reviewers agree that your paper is well written and clearly organised. They ask for minor changes before it can be accepted for publication. The reviewers have added detailed comments. Among others, changes include some terminology clarification and some additional analysis for testing the independence of various factors analysed.

Reviewer 1 ·

Basic reporting

All good.

Experimental design

All good.

Validity of the findings

All good. See review for further comments.

Additional comments

Review of Strydom et al: Fire intensity is not a major determinant…….
The paper provides valuable data on resprouting of woody shrubs in a system, thicket, whose ecology has been neglected in southern Africa (and elsewhere in the world). Snapshot surveys such as this are very useful in indicating resprouting response to an extreme event, such as the fires in this region. The study seems competent with field variables collected that seem appropriate to the questions being addressed. I’m a bit skeptical as to your ability to record small thicket plants after high severity fires which may bias the survival probability figures upwards for this size class. The analyses look good. But please add units for figures and tables recording canopy volume.
An important feature of these fires is that all the sprouting seems to have been basal since no epicormic sprouting is reported. This means that the thicket has to recover its structure from the ground up. This would allow browsing mammals greater access to the resprouts, and should also allow much more light to reach the soil layer where fynbos plants could potentially establish. Did they? Were there sufficient gaps between the thicket shrubs for colonization by shade-intolerant grasses and shrubs?
It might be worth referencing thicket responses to herbivory and fire from southern African savanna studies. As was found here, they have proven more tolerant of fire than forest species (see e.g. Charles-Dominique et al. 2015). Their sprouting behavior has been linked to bark traits, bud insulation and clonal spread below-ground. It would be intriguing to see how traits of eastern Cape thicket species compare in future studies and whether they predict species responses recorded in this study.
Other comments
Title. Too long. Too parochial. Think again for one with wider readership appeal.
Study area: For readers less familiar with the study area, are there any analyses showing differences in fire regime between fynbos and thicket?
p. 61 Replace ‘canopy’ fires with ‘crown’ fire, the more common term.
l.65. Replace coexistence with co-occurrence
l.71. coexist replace with co-occur.
l.101-102. Its quite common to have bimodal distribution of resprouting with high mortality in small plants and in old, mature plants but good survival and resprouting in intermediate ages/sizes.
l.106. How does two fires provide a wide range of fire intensities? Presumably both fires had variable fire intensity while the two regions test the geographical generality of any findings
l.124. replace ‘an excess’ with ‘in excess’.
l.131-132. What does Mucina and Rutherford’s volume on Veg of Sth Africa map the vegetation as? Cite that too.
l.150d. How many belt transects?
Figure 4. What are the units for volume in Figure 4?
Figure 2. You have to wonder if probability of survival response under high intensity was not affected by complete consumption of the smaller plants by fire – i.e. small plans were burnt out hence not sampled.
l.275-276. I think you don’t mean ‘accordingly’ but ‘in contrast’?
l.307-308. Fynbos vs thicket soils have similar chemistry e.g. in terms of OM and N. Really? I don’t think this is right. Check e.g. Figure 4 in Cowling and Potts where thicket has much higher nitrogen, Phosphorus, exchangeable cations and Carbon than fynbos. The situation seems similar to Cramer et al and Coetsee et al with feedbacks between vegetation and soil nutrients causing soil differences reinforcing veg differences supported by divergent fire regimes.
l.343. Not only fynbos. What is the effect of frequent fires on thicket expansions? Not part of your study so your recommendation on frequent (how frequent), medium intensity burns to maintain the fynbos-thicket mosaic is a reasonable guess but not really based on any data?
l.348-350. I’m puzzled by this ‘consistent’ positive effects on survival and vigour. Figure 4b from Knysna shows that shrub resprouting volume declined with increasing intensity. So not a positive effect?
A comment on shoot number versus measures of regrowth such as volume:
Shoot count = measure of the bud bank and bud survival, volume of regrowth reflects CHO reserves surviving fire obscured by, depending on the timing of the census, vigor of subsequent growth. Knysna plants showed more shoots than CSF and shoots increased from low intensity to high intensity. So high fire intensity released more buds.
Shoot volume showed different responses with high intensity fires causing reduced regrowth in Knysna relative to low intensity. So tissues may have been killed in high intensity fires resulting in less CHO reserves to resprout? The difference between the sites is apparently due to different times since burn with CSF having had longer to recover than Knysna so the link with pre-burn reserves is clouded by post-burn regrowth. Or am I mis-reading Figure 4?

Reviewer 2 ·

Basic reporting

no comment.

Experimental design

This article shows that thicket species are i) fire tolerant and ii) that this simplifies local fire management. Two issues I have relate to independence. Firstly, is fire intensity independent of plant size. The same fire could damage a small plant more than a big plant but that this would look like variable intensity. Show some analysis of this. Secondly, numbers of resprouts could be correlated with plant size and therefore number of resprouts is not an independent measure, say of plant response to fire. Show some analysis of this. Finally, I could not make out where the line is drawn between ramets and genets. Line 156 says stems > 5 cm were surveyed. Was the biggest stem per individual used or were all stems > 5 cm used and how was an individual defined. Line 223 says 85% of stems…again is this of all stems? Some stems may have died but at the individual level was there any mortality at all? This seems to be avoided in the discussion. Similarly, the value judgement that more resprouts is a positive response needs some discussion. A mild fire may have more canopy/stem survival and therefore lower basal resprouting on a particular species and this lower degree of resprouting could be a positive response to fire (i.e. greater stem survival).
Specific comments
Line 59 fire-avoiding- meaning?
Line 264 how much greater?
Line 392 reference incomplete?

Validity of the findings

no comment

·

Basic reporting

The authors have a clear and very structured writing style, I have made specific comments where there is ambiguity or grammatical errors.

The literature references and context are generally well covered, I have added specific comments below where I believe clarification is needed.

The article structure, figures and tables are presented professionally. I have made specific comments about figures below.

This is a self contained publication unit.

Experimental design

The research is original and relevant.

Research question is well defined, and the knowledge gap is clearly stated.

The investigation has been conducted with rigor. I have an issue with the data collection and subsequent analysis, I have detailed this below.

The methods are very detailed and reproducible.

Validity of the findings

The impact, novelty and management implications of this work have clearly been assessed by the authors.

The underlying data have been provided.

Conclusions are well stated and linked to the original question.

Additional comments

Broad Comments:

1) Supp Material S4 does not seem to open correctly on my device. I am not sure what the values in column 4 represent..? Did you count 808080808 resprouts on a single individual?

2) Some species in Table 1 are only sampled once or twice at the different sites. Have these species been excluded from the survival analysis? A survival rate of 100% that is calculated from one individual does not seem like a defensible assumption.

3) It would be helpful to the reader to have representative photographs of the different levels of fire intensity. This is a subjective measure, so I believe some photos in the Supp Materials would aid in reproducing this study.

4) Was the site comparison done by species? I think it would be difficult to be sure that site had no effect unless you are comparing individual species between sites. Otherwise, the community composition would have an influence on the overall survival and resprouting values.

5) The number of buds an individual has available to resprout from should increase with size. Therefore prefire-size and number of resprouts should be related. It would be interesting to see the relationship between the number of resprouts per unit of tree (controlled for size), for the different fire intensities. This would give you some relative measure of the number of active buds vs killed buds.

Specific Comments:

Line 81: Are you absolutely sure there is no research on the fire ecology? Midgley and Cowling, 1993 seems to deal with an aspect of fire ecology.

Line 174-175: See broad comment 2 above. Some species in Table 1 have one or two sampled individuals for each species. A threshold should be set to exclude species which were not sampled enough to have confidence in their resprouting response.

Line 228-229: See Charles-Dominique et al (2015) Bud protection: a key trait for species sorting in a forest–savanna mosaic.

Line 233-235: Knysna had serious drought before the fires, there may be an interaction between drought and fire intensity effects on survival and resprouting i.e. The knysna species were already stressed. (Addressed in the discussion).

Line 239-240: I would be hesitant about drawing this conclusion based on the the confidence intervals on the final panel of Figure 2.

Line 294-298: Linked to the comment above about the Charles-Dominique et al 2015 paper. As the authors state, there are considerable differences in survival rates between species. There should be considerable differences in the post-fire resprouting ability which is related to the level of insulation of the buds.

Figure 1: What does orange represent in the bottom panels?

Figure 2: A slight separation of the panels from the two sites might make this figure easier to interpret at speed.(This is a personal preference).

·

Basic reporting

This paper is well written and clearly organised. The introduction provides sufficient and reasonable background to the study and how it fits into the broader field of knowledge.
The results provided are all relevant to the hypotheses, and thus this submission is self-contained. The figures and tables presented are clear, comprehensive and appropriately described and labelled. For figure 1, I would suggest adding a legend to the satellite images to be clear that they are showing the extent of the fires rather than the extent of the vegetation type.

My only major comment is regarding the use of the terms fire intensity vs fire severity. See Keeley JE (2009) Fire intensity, fire severity and burn severity: a brief review and suggested usage. IJWF 18: 116-126. What the authors have measured here is technically fire severity (i.e. vegetation damage) rather than fire intensity (e.g. energy output, fire temperatures). Of course, these two fire characteristics are often correlated, but for accuracy, I would suggest considering if fire severity is the more appropriate term to use in many places throughout this manuscript (including in the title).

Experimental design

This research fits well within the Aims and Scope of PeerJ. The research questions are clear and meaningful and identify a fundamental knowledge gap. The experimental design is excellent and described clearly. The investigation appears to have been conducted rigorously and to a high standard.

Validity of the findings

The raw data have been made available in the supplementary material. I suggest adding metadata for these raw data, especially units for the size and volume measurements.
The conclusions stated are appropriate and clearly supported by the results presented. A few comments below:

Lines 332-334: Presumably, small fynbos shrubs are also vulnerable to frequent fires –resprouters that are too small to survive and obligate seeders that have not yet produced sufficient seed. Therefore, a frequent fire regime may have unintended consequences for fynbos shrub species. It is probably worth commenting on the comparative survival/reproductive effort of small shrubs of fynbos vs thicket species and what this could mean for their response to frequent fires – even if this is unknown, it is worth mentioning that.

Lines 341-342: What is the justification for suggesting a moderate intensity fire regime? Presumably this is because fynbos vegetation is fire dependent, i.e. many species require fire to promote seedling recruitment, and therefore a low intensity fire regime may not provide germination/seed release cues. Maybe worth briefly mentioning this reason at the end of this sentence, e.g. “... and provide fire regimes appropriate for fire-dependent fynbos species.” Also relevant to this sentence and my previous comment – you should include at least a rough estimate of what frequent fire regimes are, e.g. 5-10 years?

Additional comments

This is a well presented and well written manuscript that I found easy to read. It clearly helps develop key ecological knowledge of the thicket-fynbos ecotone, with clear implications for fire and land management. Some minor comments below:
Lines 56 – 58: to help with the flow of this sentence, consider moving “of both fire-dependent and fire-avoiding biomes” before “in the same mesoclimate and geology”
Lines 105 – 106: suggest adding “and under different weather and climatic conditions” after “that burnt at different times”
Line 118: Should this refer to Table S1 rather than Fig. S1?
Line 131: add a comma after ‘classification’
Line 136-137: Where species names are mentioned, I would suggest including plant families for the benefit of an international audience
Line 138: remove comma after ‘mesophyllous’
Line 153: add closing bracket after ‘Cape St Francis site’
Lines 155-157: This sentence is a bit awkward. Consider splitting it in two, i.e. “Within transects, we measured each shrub with a stem diameter > 5 cm. Measurements included...” Also clarify if this is basal stem diameter (10 cm above the ground).
Lines 157-161: I would suggest explicitly stating that this measurement is called Firebase. This will help the reader to easily interpret the figures presented.
Line 188: weigh -> weight

Also, check for spelling errors in the reference list, e.g.:
Line 355: statstical -> statistical; indentification -> identification
Line 364: reguirments -> requirements
Line 379: persistance -> persistence
Line 380: disperesed -> dispersed
Line 382: succesion -> succession

---

## Round 0.2 · accepted · Accept

The reviewer and I agree that the changes required since the first version of the work have been made satisfactorily. The work is now suitable for publication in PeerJ.

·

Basic reporting

No comment

Experimental design

No comment

Validity of the findings

No comment

Additional comments

The authors have addressed the reviewers' comments sufficiently.